# Quantifying and Optimizing Global Faithfulness in Persona-driven Role-playing

**Letian Peng, Jingbo Shang**[*]
Department of Computer Science
University of California, San Diego
{lepeng, jshang}@ucsd.edu

## Abstract

Persona-driven role-playing (PRP) aims to build AI characters that can respond to user queries by faithfully sticking with *all* (factual) statements in persona documents. Unfortunately, existing faithfulness criteria for PRP are limited to coarse-grained LLM-based scoring without a clear definition or formulation. This paper presents a pioneering exploration to quantify PRP faithfulness evaluation as a fine-grained and explainable criterion, which also serves as a reliable reference for faithfulness optimization. Our criterion first discriminates persona statements into *active* and *passive* constraints by identifying the query-statement relevance. Then, we incorporate all constraints following the principle that the AI character's response should be (a) entailed by active (relevant) constraints and (b) not contradicted by passive (irrelevant) constraints. We translate this principle mathematically into a novel Active-Passive-Constraint (APC) score, a constraint-wise sum of statement-to-response natural language inference (NLI) scores weighted by constraint-query relevance scores. In practice, we build the APC scoring system by symbolically distilling small NLI and relevance discriminators ($\sim$300M parameters) from GPT-4 for efficiency, and both show high consistency with GPT-4's discrimination. We validate the quality of the APC score against human evaluation based on example personas with tens of statements, and the results show a high correlation. As the APC score could faithfully reflect the PRP quality, we further leverage it as a reward system in direct preference optimization (DPO) for better AI characters. Our experiments offer a fine-grained and explainable comparison between existing PRP techniques, revealing their advantages and limitations. We further find APC-based DPO to be one of the most competitive techniques for sticking with all constraints and can be well incorporated with other techniques. We then extend the scale of the experiments to real persons with hundreds of statements and reach a consistent conclusion. Finally, we provide comprehensive analyses and case studies to support the effectiveness of APC evaluation and APC-based DPO. [2]

## 1 Introduction

Role-playing (Han et al., 2022; Li et al., 2023; Yan et al., 2023; Bianchi et al., 2024; Yu et al., 2024; Tao et al., 2024) is a newborn and trending natural language processing field, emerging from the proficiency of large language models (LLMs) (Brown et al., 2020; OpenAI, 2023; Touvron et al., 2023a,b; Mesnard et al., 2024) in human interaction. Role-playing customized AI characters, which are useful for providing emotional value (Zhang et al., 2024), developing video games (Hu et al., 2024), or even realizing the metaverse (Zhou, 2023; Yue et al., 2024). Persona-driven role-playing

---

[*] Corresponding author.

[2] Code, Dataset, Demo: https://github.com/KomeijiForce/Active_Passive_Constraint_Koishiday_2024

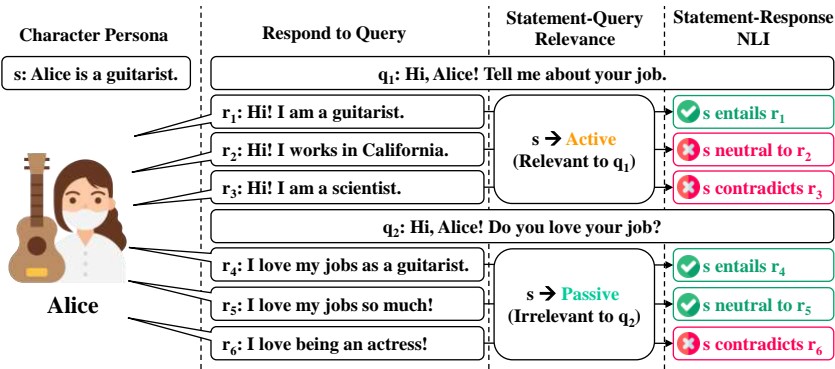

Figure 1: A presentation of the alignment between APC and human's view on PRP faithfulness.

(PRP) (Wang et al., 2023a,b; Shao et al., 2023; Xu et al., 2024) uses only persona statements to efficiently build the AI character without dialogues or scripts, which is extremely useful for real-world applications as few characters have sufficient or accessible dialogues for training.

As the persona statements are the only input in PRP, being faithful to them becomes one of the most crucial objectives for this task. Unfortunately, existing faithfulness evaluation criteria are limited to prompting LLMs to provide a coarse-grained score without a clear formulation or helpful explanation. Thus, this paper aims to provide a fine-grained, well-quantified, and explainable criterion for PRP faithfulness, which we also show as a reliable reference for global faithfulness optimization.

Our criterion views PRP as a constraint satisfaction problem (CSP) (Brailsford et al., 1999), and the whole persona information as a global constraint for the response to satisfy. Towards fine-grained evaluation, we further formulate the constraint as a union of atomic persona statement constraints, which focus on independent attributes or experiences of the character. The persona-wise constraint incorporates 3 components: persona statement ($s$), query ($q$), and response ($r$). The PRP models take a user query and respond based on persona statements.

Our key insights are 1) the statement-to-response constraint depends on query-statement relevance and 2) the statement-to-response constraint can be formalized as statement-to-response natural language inference (NLI). (Bowman et al., 2015) The constraint becomes *active* when the query is relevant to the persona statement, constraining the response to be entailed by the persona statement. The constraint becomes *passive* when the query is irrelevant to the persona statement, reducing the constraint to only not being contradicted by the persona statement. We present a possible PRP instance in Figure 1 to show how our definition is consistent with human's view on PRP faithfulness. As $q_1$ is relevant to $s$, $s$ becomes active and constrains the character "Alice" to incorporate the information in $s$ to her response. For irrelevant $q_2$, the constraint of $s$ becomes passive and is relaxed to only not incorporating information contradicting $s$.

We further develop a scoring system to quantify APC, making it more appropriate for evaluating practical PRP methods. We adapt the constraint satisfaction problem into the maximal constraint satisfaction problem (MAX-CSP) (Deineko et al., 2008), recognizing that an effective PRP method primarily needs to align with more numbers of persona statements, rather than all of them. Thus, the quantified APC score sums up the satisfaction probability of the response to each persona statement, representing the expected number of satisfied constraints. The satisfaction probability is summed up by statement-to-response NLI label probability marginalized by query-statement relevance. We also regularize the APC score to ΔAPC score with a minuend equal to the reward gained by a PRP system that permanently gives a neutral response. The regularization makes the absolute value more straightforward to reflect faithfulness, representing the expected number of entailed active persona statements (active reward) subtracted by the expected number of contradicted passive persona statements (passive penalty). In practice, the probabilities are efficiently assigned by small discriminators based on DeBERTa-V3 (He et al., 2021) ($\sim$300M parameters) symbolically distilled from the state-of-the-art LLM, GPT-4 (OpenAI, 2023) with $\sim 90\%$ accuracy.

With the (Δ)APC score, we can reveal the advantages and limitations of existing PRP methods. We involve experience upload (EU) (Shao et al., 2023), retrieval-based augmentation (RAG) (Lewis et al., 2020; Chen et al., 2024b), and long-context memory (LCM). We handcraft 3 original characters with small-scale persona statements (8, 19, 30) and free from data contamination (Magar & Schwartz,

2022) in the pre-training of LLMs. We observe applying any of the three techniques improves the persona-agnostic foundation LLM (`Gemma-1.1-7b-it`), indicating their benefits to PRP. However, our experiments also confirm that their limitations are significant. EU constructs character experiences based on each persona statement, but these often meet only some constraints and sometimes even violate them, whether actively or passively. RAG adheres more closely to the given personas, incorporating more relevant statements, though it still sometimes misses passive constraints. LCM, on the other hand, loads the entire persona into the context in hopes that the LLM will effectively utilize all persona statements. Our experiment shows that as the number of persona statements increases, LCM's performance deteriorates compared to RAG, confirming findings about limitations in LLMs' handling of long contexts as discussed in Liu et al. (2024b).

Furthermore, we discover the APC score to be a reliable reward for direct preference optimization (DPO) (Rafailov et al., 2023) to strengthen the faithfulness of PRP methods. We use APC and human evaluation to verify the effectiveness of DPO, which benefits the satisfaction of both active and negative constraints. We extend the experiments for evaluation and DPO above to complicated famous figures with $77 \sim 599$ persona statements, further verifying the reached conclusions.

Finally, we launch case studies toward a specific analysis of the insights obtained by APC score-based evaluation and the benefit gained from APC-based DPO. We also showcase how we can explain the detected constraint violation by tracing back and strengthening extra constraints like protective experience by persona statements. Our contribution is three-fold,

- We propose the first formal definition of AI character's global faithfulness and formulate it as a constraint satisfaction problem. The constraint is further quantified as the APC score, which is human-consistent and the first quantified evaluation for AI characters.
- We evaluate potential PRP techniques, EU, RAG, and LCM by APC score, which reveals their properties on active and passive constraints.
- We find APC-based DPO to be one of the most competitive techniques to improve the global faithfulness of AI characters and cooperate well with other methods.

## 2 Related Works

With the emergence of the high capability of LLMs in interaction with humans, role-playing AI has attracted lots of attention from both academia (Shanahan et al., 2023) and industry[3]. The difference between role-playing and normal agents is the demand of following a constant persona. The main aim of role-playing includes personalizing the agent for the user preference (Jang et al., 2023) and bringing virtual characters to the real world (Li et al., 2023; Tao et al., 2024). Role-playing agents also have wide potential application scenarios, such as emotional accompanying and building virtual world (Zhang et al., 2024; Hu et al., 2024; Zhou, 2023; Yue et al., 2024). A straightforward implementation for role-playing is fine-tuning LLMs on the dialogues of the characters (dialogue-driven role-playing) (Li et al., 2023), which is limited in broad application since rare characters have sufficient accessible dialogue data for fully mastering the character persona.

**Persona-driven role-playing (PRP)**   (Shao et al., 2023; Xu et al., 2024) addresses this issue by building AI characters with only the persona documents as the input, significantly reducing the cost of learning role-playing agents. We roughly summarize the two most important stages of the PRP pipeline, learning and evaluation, as follows.

**Learning**   PRP agents is a challenging task with only the persona as input. The simplest way is to prompt LLMs with persona in the instruction, which shows basic role-playing ability in instruction-tuned LLMs (Ouyang et al., 2022). Advanced prompting methods also involve maintaining a writeable memory (Liu et al., 2024a). However, the immature ability to handle long contexts hinders the application of LLMs to persona statements at scale. Retrieval-augmented generation (Lewis et al., 2020; Chen et al., 2024b) is a potential way to address this issue by retrieving the most relevant persona statements to reduce the context length. Besides incorporating persona information into the prompt, Shao et al. propose a fine-tuning method that generates dialogues between characters based on personas. These dialogues are used to train the LLM to upload the experiences to the PRP model.

---

[3]https://character.ai/

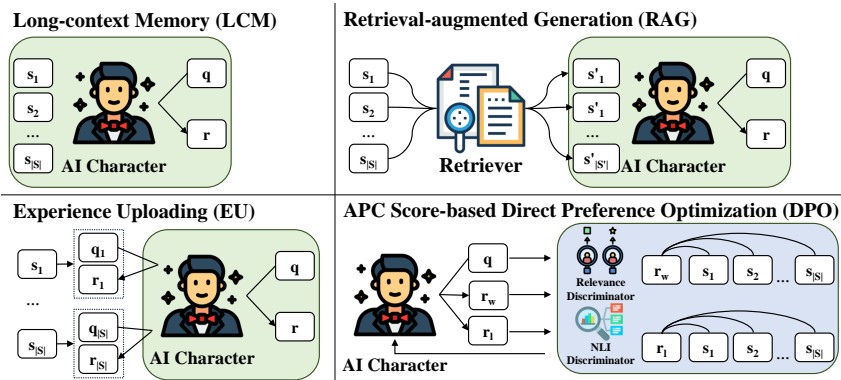

Figure 2: An overview of different PRP methods.

**Evaluation** is a crucial aspect of PRP systems. Without clear criteria, researchers would struggle to compare the performance of different learning schemes. Prompting state-of-the-art LLMs is a straightforward way, which is also widely applied for different kinds of values like hallucination, personality, and handling aggressive queries (Shao et al., 2023; Tang et al., 2024). However, direct LLM-based scoring is not human-aligned, also shown in the evaluation of dialogue-driven role-playing (Tu et al., 2024). Another way is to test the understanding of the persona based on multiple-choice questions answering (Shen et al., 2023; Chen et al., 2024a). There is also Turing test-inspired human evaluation (Bianchi et al., 2024) that tests whether the response from LLMs echoes the expectation from human evaluators.

Unfortunately, these evaluation methods for PRP are either vague or indirect. Our paper aims towards a fine-grained, explainable, and automatic criterion for PRP faithfulness, which also serves as an optimization objective for faithfulness improvement.

## 3 Preliminary

### 3.1 Persona-driven Role-Playing

A persona-driven role-playing (PRP) agent (AI character) is defined as a function $f(\cdot)$ that $r \sim f(q|S)$, which generates a response $r$ to a query $q$ (including the history in multi-turn interactions), referring to persona statements $S = [s_1, s_2, \cdots, s_{|S|}]$. Ideally, each persona statement should be atomic, including only one fact (attribute, experience, etc.) about the character. Existing PRP agents are mostly based on LLMs, denoted as $f_{\text{LLM}}(\cdot)$, taking a prompt as the input and outputs a response.

### 3.2 In-context PRP

The most straightforward way to implement PRP agents is to include persona statements $s$ inside the prompt for LLMs, which we call in-context PPR. Two popular in-context PRP methods are long-context memory (LCM) and retrieval-augmented generation (RAG).

**Long-context Memory** directly includes all persona statements ($S$) in the prompt and asks the LLM to respond, $r \sim f_{\text{LLM}}(S \oplus q)$. Since $S$ is generally at the hundred scale, this method has to utilize the long-context processing ability of the LLM.

**Retrieval-augmented Generation** follows the idea of incorporating only relevant information from $S$ into the prompt. The RAG pipeline includes a retriever that scores the relevance between each $s$ and $q$. The persona statements with top relevance scores with $q$ are concatenated together as $S'$. Finally, $S'$ is incorporated into the prompt for response generation, $r \sim f_{\text{LLM}}(S' \oplus q)$

### 3.3 Experience Upload

Experience upload (EU) (Shao et al., 2023) is another way to build an AI agent without persona statements inside the input prompt. For each persona statement $s$, EU prompts the LLM to generate

$(q, r)$ pairs that $q$ is generally relevant to $s$ and $r$ is faithful to $s$. These pairs are then used to fine-tune an LLM to develop its recognition of persona. On the role-playing stage, the LLM only takes the query as input, $r \sim f_{\text{LLM}}(q)$.

## 4  Active-Passive-Constraint

### 4.1  Definition and Formulation

We first recall the high-level idea of APC mentioned in the introduction that we aim to formulate faithful PRP as a constraint satisfaction problem (CSP). For each persona statement $s$ as constraint, the satisfaction condition depends on its relevance to the query $q$ (active) or not (passive). We introduce a Boolean function $g(\cdot)$ to represent this status, $g(s, q)$ returns 1 when $s, q$ are relevant and returns 0 for irrelevance. When the constraint is active ($g(s, q) = 1$), the response $r$ is constrained to be entailed by $s$, denoted as $s \models r$ (MacCartney & Manning, 2014). When the constraint is passive ($g(s, q) = 0$) in natural language inference (NLI), the constraint for $r$ is released to only not being contradicted by $s$, denoted as $s \not\models \neg r$. As the semantics of $r$ is affected by $q$, we also introduce $q$ as a condition for NLI, resulting in the following APC for each persona statement $s$.

$$\text{APC}(q, r | s) = (g(s, q) \land (s \models r | q)) \lor (\neg g(s, q) \land (s \not\models \neg r | q)) \tag{1}$$

Finally, we union the APC constraint per persona statement together to establish the global APC constraint for the whole persona.

$$\text{APC}(q, r | S) = \wedge_{i=1}^{|S|} \text{APC}(q, r | s_i) = \wedge_{i=1}^{|S|} [(g(s_i, q) \land (s_i \models r | q)) \lor (\neg g(s_i, q) \land (s_i \not\models \neg r | q))] \tag{2}$$

### 4.2  Mathematical Quantification

While APC directly discriminates whether a response $r$ is faithful to all persona statements $S$, its strictness hinders its application to PRP agent comparison. Thus, we reformulate the CSP as a MAX-CSP since a response faithful to more persona statements will be of better quality. The metric, APC score ($V_{\text{APC}}(\cdot)$) counts the number of constraints satisfied by the response. To further fine-granularize the metric, we introduce $P_{\text{APC}}(\cdot)$ evaluating the probability of each constraint being satisfied.

$$V_{\text{APC}}(q, r | S) = \#_{i=1, \cdots, |S|}[\text{APC}(q, r | s_i)] = \sum_{i=1}^{|S|} P_{\text{APC}}(q, r | s_i) \tag{3}$$

The $P_{\text{APC}}(q, r | s_i)$ is marginalized by the probability of statement-query relevance, which is represented by two probabilistic evaluators $P_g(\cdot)$ for statement-query relevance and $P_h(\cdot)$ for statement-to-response NLI.

$$P_{\text{APC}}(q, r | s_i) = (P_g(s_i, q) P_h(s_i \models r | q)) + (1 - P_g(s_i, q)) P_h(s_i \not\models \neg r | q) \tag{4}$$

Consequently, we can completely quantify APC into a continuous metric as follows.

$$V_{\text{APC}}(q, r | S) = \sum_{i=1}^{|S|} [(P_g(s_i, q) P_h(s_i \models r | q)) + (1 - P_g(s_i, q)) P_h(s_i \not\models \neg r | q)] \tag{5}$$

**Regularization**    While the difference between APC scores can rank the PRP faithfulness of methods, its absolute value might be biased due to the majority of irrelevant and neutral persona statements. Thus, we introduce **ΔAPC score** to regularize the absolute value by reducing the APC score gained by a PRP algorithm that always outputs responses neutral to any persona statement.

$$\Delta V_{\text{APC}}(q, r | S) = V_{\text{APC}}(q, r | S) - \sum_{i=1}^{|S|} (1 - P_g(s_i, q)) \tag{6}$$

As the minuend is independent of the evaluated PRP method, **ΔAPC score** still discriminates the PRP faithfulness of methods. The value of **ΔAPC score** reflects the difference between the expected entailed active constraint number (active reward) and the expected contradicted passive constraint number (passive penalty), which offers a more straightforward view of the PRP faithfulness.

### 4.3  Weakness of PRP Methods from APC's View

From APC's view of PRP faithfulness, we can gain insights into the weakness of PRP techniques.

- **EU** creates $(q, r)$ pairs based on each $s$ to fine-tune a LLM. While the pair $(q, r)$ generally meets $\text{APC}(q, r|s)$ by satisfying $g(s, q) \land (s \models r)$, it fails to meet other constraints because they are not included in the prompting process. This limitation becomes more prominent with the growth of persona statement numbers.
- **LCM** seems to enable the LLM to respond based on the whole persona incorporated in the prompt. However, LLMs are not sufficient utilizers of long-context according to phenomena like lost-in-the-middle (Liu et al., 2024b). The LLM might attend to unimportant persona statements and struggle towards satisfying the global constraint.
- **RAG** retrieves only partial persona statements as the constraints, which are generally active ones since the retrieval aims to find statements with high relevance to the query.

### 4.4 APC-based Direct Preference Optimization

Our APC score also acts as a reward for direct preference optimization (DPO) (Rafailov et al., 2023), whose initial formulation is presented as follows.

$$\mathcal{L}_{\text{DPO}}(\pi_\theta, \pi_{\text{ref}}) = -\mathbb{E}_{(x, y_w, y_l) \sim \mathcal{D}} \left[ \log \sigma \left( \beta \log \frac{\pi_\theta(y_w|x)}{\pi_{\text{ref}}(y_w|x)} - \beta \log \frac{\pi_\theta(y_l|x)}{\pi_{\text{ref}}(y_l|x)} \right) \right] \quad (7)$$

where $y_w$ is more preferred than $y_l$ referencing to a reward model $\pi_{\text{ref}}(\cdot)$, the DPO loss uses the reward value to $y_w, y_l$ to align the LLM's preference with the reward model. Following the formulation of the APC score, there are two reward models, $\pi_{(a)}, \pi_{(p)}$, for active and passive constraints.

$$\pi_{\text{ref}}(r|g(s, q)) = \pi_{(a)}(r|q, s_i) = P_h(q \models r|s_i); \pi_{\text{ref}}(r|\neg g(s, q)) = \pi_{(p)}(r|q, s_i) = P_h(q \not\models \neg r|s_i). \quad (8)$$

We combine the $\mathcal{L}_{\text{DPO}}$ for $\pi_{(a)}$ and $\pi_{(p)}$ depending on $P_g(s_i, q)$ to formulate the final loss. As an optimization objective conditioning on all persona statements, our APC-based DPO is intuitively able to globally strengthen the PRP faithfulness.

$$\mathcal{L}_{\text{APC}}(\pi_\theta, \pi_{(a)}, \pi_{(p)}) = \sum_{i=1}^{|S|} P_g(s_i, q) \mathcal{L}_{\text{DPO}}(\pi_\theta, \pi_{(a)}) + (1 - P_g(s_i, q)) \mathcal{L}_{\text{DPO}}(\pi_\theta, \pi_{(p)}) \quad (9)$$

## 5 Experiments

### 5.1 Implementation Details

**Evaluation** We follow Shao et al. (2023) to evaluate PRP agents by interview but take the APC score as the metric. We implement the APC score criterion by symbolically distilling from the state-of-the-art LLM, GPT-4 (OpenAI, 2023) and report the regularized **ΔAPC score**. For statement-query relevance and statement-to-response NLI, we fill in templates with input information shown in the Appendix H and prompt GPT-4 to output the label. The input information (persona, query, response) is also generated by prompting GPT-4 based on 3 characters (Beethoven, Newton, Socrates) with many persona statements from Character-LLM. We got $8.4K$ data for statement-query relevance and $18.9K$ data for statement-to-response NLI, which are used to fine-tune a state-of-the-art discriminator DeBERTa-V3 ($\sim 300M$ parameters) (He et al., 2021) for efficiency. We use $80\%/20\%$ train/test split and observe a high ($\sim 90\%$) accuracy referencing GPT-4's labels, which guarantees a high capability of the distilled discriminator. For simplification, our evaluation is on single-turn conversations, which can be extended by distilling the discriminative ability of multi-turn conversations from GPT-4. More details about the distillation can be found in the Appendix C. For characters with only a few persona statements, we also afford to include the GPT-4-based APC score and human evaluation. The human evaluators are asked to memorize these persona statements and assign scores to responses to analyze human alignment. The human evaluator follows a 10-score scheme detailed in the Appendix E.

**Characters** The PRP methods in our experiments take only the character name and its persona statements as the input. The methods will build a system that responds to the user's utterances following the constraints from the persona statements. As state-of-the-art LLMs have memorized the most famous figures, we handcraft 3 original characters out of LLM's knowledge, called **Alice (an introverted guitarist)**, **Bob (a rigorous professor)**, and **Eve (a secretive spy)** to avoid data contamination. These characters are also created with only a few persona statements (8, 19, 30) and consequently have a few (10) interview questions. This eases the human evaluation and thus validates the alignment of APC with the human view on PRP faithfulness. We also include the 6 characters (Spartacus, Hermione, Voldemort, Cleopatra, Caesar, Martin Luther King) not used to

| Character | Alice | | | Bob | | | Eve | | |
|---|---|---|---|---|---|---|---|---|---|
| #Statement | 8 | | | 19 | | | 30 | | |
| Evaluator | ΔAPC | | Human | ΔAPC | | Human | ΔAPC | | Human |
| | DeB | GPT-4 | | DeB | GPT-4 | | DeB | GPT-4 | |
| **w/o CPO** Gemma-7B | 0.7 | 0.3 | 1.8 | 1.1 | 0.4 | 1.8 | 0.7 | −0.2 | 2.0 |
| EU | 2.6 | 1.1 | 6.4 | 3.4 | 1.1 | 6.2 | 3.6 | 0.7 | 4.6 |
| LCM | 2.6 | 1.4 | 6.8 | 4.5 | 2.2 | 7.2 | 3.9 | 0.7 | 5.0 |
| RAG | 2.8 | 1.8 | 6.8 | 4.0 | 1.7 | 6.8 | 4.8 | 2.4 | 5.8 |
| **w/ CPO** EU | 2.7 (+0.1) | 1.4 (+0.3) | 6.8 (+0.4) | 3.8 (+0.4) | 1.8 (+0.7) | 6.8 (+0.6) | 3.9 (+0.3) | 0.9 (+0.2) | 5.2 (+0.6) |
| LCM | 2.8 (+0.2) | **2.2** (+0.8) | **7.6** (+0.8) | **5.3** (+0.8) | 2.5 (+0.3) | 7.8 (+0.6) | 5.1 (+1.2) | 3.3 (+2.6) | 6.6 (+1.6) |
| RAG | **2.9** (+0.1) | **2.2** (+0.4) | **7.6** (+0.8) | 5.2 (+1.2) | **3.8** (+2.1) | **8.2** (+1.2) | **5.8** (+1.0) | **4.2** (+1.8) | **7.0** (+1.2) |

Table 1: PRP Faithfulness Evaluation on simple and data contamination-free characters. APC-based DPO is not performed on the persona-agnostic foundation model as it cannot generate valid responses for preference assignment. **CPO:** Abbreviation of our APC-based D**PO**

build the evaluator, which have many persona statements to evaluate the faithfulness of PRP methods at scale. Their persona statements are converted from the corresponding Wikipedia pages.

## 5.2 Compared Methods

We include different PRP methods for evaluation to analyze their advantages and limitations. All methods, except prompting closed-source LLMs, use Gemma (`Gemma-1.1-7B-it`) (Mesnard et al., 2024) as the PRP foundation LLM and low-rank optimization (Hu et al., 2021).

- **Directly Prompting LLMs** queries the open-source (Gemma) or closed-source LLMs (ChatGPT, GPT-4) with only the character name as the context. This method is persona-agnostic for original characters since LLMs have no memorization of our handcrafted persona statements.
- **Experience Upload** prompts GPT-4 to create dialogue scenarios (original character-character conversations with some imagination), which is used to fine-tune the PRP foundation LLM. Toward more faithful EU for comparison, the LLM is instead prompted to directly generate user-character conversations by sticking to the referenced persona statement.
- **Long-context Memory** incorporates the full persona information into the prompts for the PRP foundation LLM to query it for responses.
- **Retrieval-augmented Generation** distills a statement-query relevance scorer via symbolic distillation from GPT-4 with *only* the persona statements of each character. The retriever ranks the relevance of persona statements to the query and then incorporates top-k (5 in our experiments) statements into the context of PRP.
- **APC-based Direct Preference Optimization** assigns preference to sampled responses from PRP methods by APC score. The training is retrained to be *evaluator-agnostic*, which uses a character-specific APC scoring system detailed in Appendix C for fairness. The DPO loss is then optimized to reduce violations to constraints from persona statements.

The setup of hyperparameters can be found in the Appendix D for reproduction. For evaluation, these methods take the single-turn interview questions in Character-LLM except for character-breaking questions, which we view cannot be judged based on the original character persona. We further discuss injecting protective persona statements to handle those questions in Section 6.3.

## 5.3 PRP as Simple Original Characters

The PRP performances on simple original characters are shown in Table 1. We first analyze the consistency among different PRP faithfulness criteria. Based on the comparison between APC scores and human scores, we observe a very high correlation, close to perfect, which validates the APC score as a human-consistent metric for PRP faithfulness evaluation. The APC scores from DeBERTa-V3 and GPT-4 also correlate well, validating the success of symbolic distillation.

Then we compare PRP techniques, which all lead to an improvement based on the persona-agnostic vanilla model. Among PRP techniques, EU performs the worst, consistent with the APC-based

| Character | Spartacus | Hermione | Voldemort | Cleopatra | Caesar | MLK | Average |
|---|---|---|---|---|---|---|---|
| #Statement | 77 | 146 | 201 | 374 | 498 | 599 | |
| **GPT** ChatGPT | 2.6 | 1.4 | −3.0 | −0.6 | 1.7 | 11.9 | 2.3 |
| GPT-4 | 2.5 | 2.5 | −2.0 | 1.5 | 5.1 | 15.1 | 4.1 |
| **w/o CPO** Gemma-7B | 2.3 | 2.3 | 1.4 | 2.4 | 3.5 | 9.6 | 3.5 |
| EU | 0.9 | −1.1 | −5.5 | −3.2 | −1.6 | 6.8 | −0.7 |
| RAG | **3.6** | 3.0 | 3.0 | **3.4** | 5.4 | 16.3 | 5.7 |
| **w/ CPO** Gemma-7B | 2.9 | 3.2 | 4.8 | 2.0 | 3.1 | 18.1 | 5.6 |
| EU | 2.2 | 0.8 | −0.7 | −0.2 | −1.3 | 6.9 | 0.2 |
| RAG | 3.4 | **3.9** | **5.0** | 3.0 | **6.4** | **19.9** | **6.9** |

Table 2: PRP Faithfulness Evaluation (ΔAPC score) on characters with persona statements at scale.

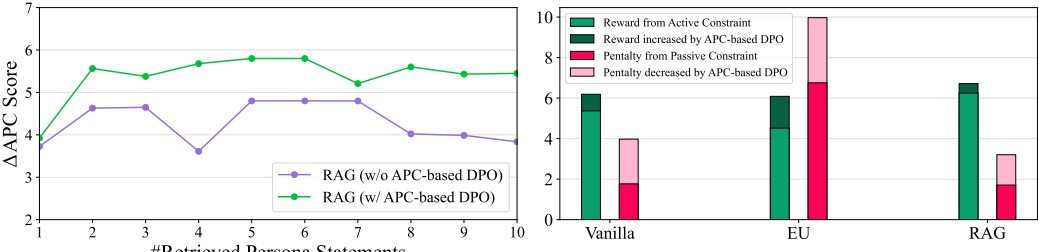

Figure 3: **Left:** The scaling rule of the number of in-context persona statements with ΔAPC scores. **Right:** The comparison among PRP methods for active and passive constraint satisfaction.

hypothesis that the generated memory for uploading will violate some constraints. We further specifically showcase this violation in Section 6.2. Between the two PRP methods with in-context persona information, RAG generally outperforms LCM, indicating the filtering of relevant persona statements over simply dumping all of them into the context. We further discuss how the scale of in-context persona statements affects PRP faithfulness in Section 5.5.

Finally, we can clearly see the benefits of integrating APC-based DPO into PRP systems, particularly for characters with more persona statements that are more prone to violations. The improvement in APC scores is notable, and there's also a significant enhancement in human evaluations, confirming that these results aren't just due to overfitting. In Section 6.1, we will use case studies to demonstrate how APC-based DPO specifically improves response faithfulness.

## 5.4 PRP as Complicated Famous Figures

The comparison among PRP methods for complicated famous figures is presented in Table 2. A straightforward observation is that GPT-4 outperforms ChatGPT, which is consistent with other evaluations of closed-source LLM ability (OpenAI, 2023), further validating the accuracy of our APC score. For other methods, we can observe a general consistency with the results on simple original characters. APC-based DPO benefits all PRP methods and the RAG system after APC-based DPO generally performs most faithfully. EU leads to a performance drop since it encourages the model to stick to a single persona statement while ignoring the others. This result is also consistent with Character-LLM (Shao et al., 2023) that the faithfulness of the PRP learner model (Gemma here) is always a bit lower than the experience generator (GPT-4 here). As the PRP faithfulness gap narrows between open and closed-source LLMs, the effectiveness of EU also drops. Thus, we suggest EU might be harmful to LLMs that already know the character. Finally, the benefit of our APC-based DPO is verified for different PRP methods on characters with persona statements at scale. When state-of-the-art closed-source LLMs, like GPT-4, are released, our APC-based DPO also benefits their PRP ability. We continue the discussion on the full APC scores in Appendix G.

## 5.5 Property Analysis of PRP Methods

**Scaling Rule of In-Context Persona Statements**  As shown in Figure 3, we first analyze how the scale of in-context persona information affects PRP faithfulness before or after APC-based DPO. We experiment on PRP as Eve for instance. The most effective in-context persona statement number is $5 \sim 7$, and faithfulness drops with a longer context, showing the reason LCM is outperformed by RAG. Before APC-based DPO, a longer context ($8 \sim 10$ persona statements) is even outperformed

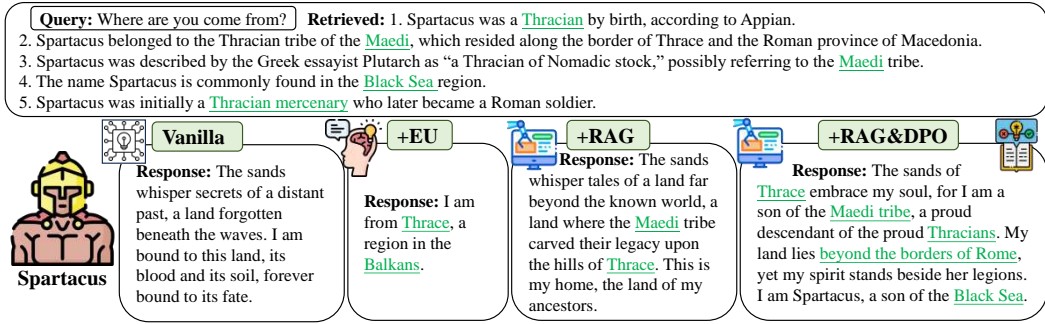

Figure 4: Case studies of different PRP techniques.

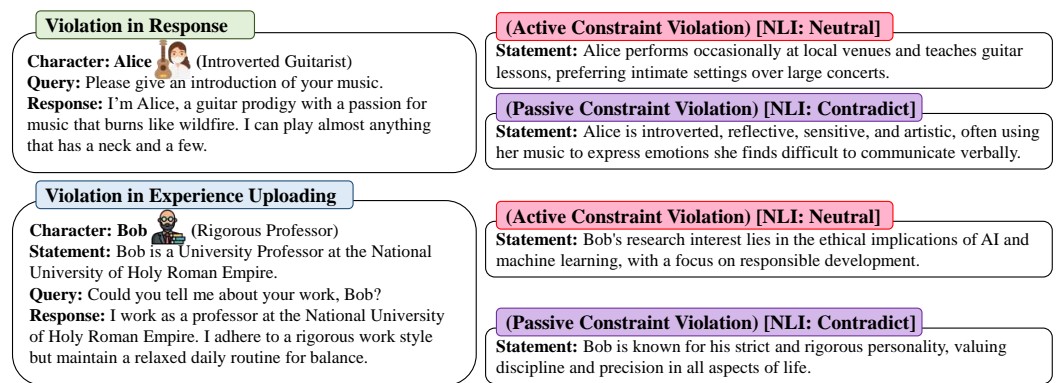

Figure 5: Case studies of violations in response and experience upload.

by very short contexts (2 ~ 3 persona statements). After DPO, faithfulness drops in longer contexts and becomes less prominent, indicating the robustness improvement of LCM from APC-based DPO.

**Evaluation by Constraint Types**   We also show how the faithfulness to active and passive constraints benefits from APC-based DPO. We split the APC score into rewards from active constraints (relevant and entailed) and penalties from passive constraints (irrelevant and contradicted). We use PRP as Voldemort for instance. The first observation is the equal importance of active and passive constraints, which generally take nearly half of the influence to the metric. Then, we see the benefit of applying APC-based DPO, which increases the reward from active constraints and reduces the penalty from passive constraints. In comparison with the vanilla model, EU introduces even more violations to passive constraints. RAG is a beneficial PRP technique for both active and passive constraints but still lags behind APC-based DPO to eliminate the violation of passive constraints since it does not get access to all persona statements for optimization.

## 6   Case Study

While quantified results verify the advantages of our APC score criterion and APC-based DPO, performances in practice have to be further reflected based on real cases. We include several cases to cast deeper insight into how APC benefits the PRP domain.

### 6.1   Real Case Analysis

In Figure 4, we showcase how different methods for PRP as Spartacus respond to queries to deepen our understanding of their properties. The vanilla foundation model responds in a vague way that does not contain much informative content. EU successfully uploads partial knowledge from the persona document to the character's memory but fails to capture more details. RAG performs similarly, which only incorporates partial information into the response and includes some ambiguity like describing

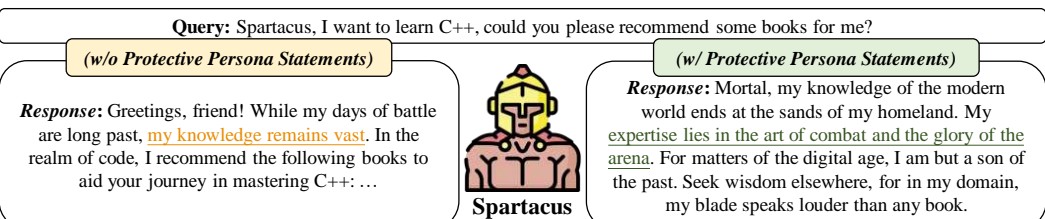

Figure 6: Effect of protective persona statements on PRP.

the hometown as "a land far beyond the known world". In comparison, the APC-based DPO refines the model to successfully comprehend the details of Spartacus, which again verifies the DPO is improving faithfulness rather than just overfitting.

## 6.2 Violation Detection

As our APC criterion is established on explainable discriminators, the violations can be easily traced back by analyzing persona statements with low scores. Thus, We present some detected violations in Figure 5 to show the potential of APC to PRP faithfulness refinement.

**Violation in Response**   We show the violations of a response from the PRP method (specifically EU for Alice). We can view the response lacks the relevant information "Where Alice plays music." and is contradicted by the fact that "Alice is introverted." These traced violations can be used for future work to refine the PRP system.

**Violation in Experience Upload**   We also use APC to specifically explain why EU sometimes uploads hallucinated information to PRP models. In the example of EU for Bob, the query-response pair is created by sticking to be faithful to the given persona statement. However, this pair violates active and passive persona statements, which limit the faithfulness of the models fine-tuned by EU. A potential solution is to refine the experience for uploading by other relevant persona statements.

## 6.3 Protective Persona Statement

Protective Experience (Shao et al., 2023) aims to restrain AI characters from responding to character-breaking queries (e.g., *"Could you recommend some C++ books?"*). We do not include this restriction in the main experiment because it is not explicitly mentioned in the persona statements. Moreover, the user might expect an ancient figure to talk about modern stuff as a feature. Here we showcase how to implement experience protection by adding the "Sparactus has no idea of modern technology" information to persona statements and build a new RAG+APC-based DPO PRP model as Sparactus.

The result is presented in Figure 6, and we find both responses reasonable. The left one without protective persona statements role-plays as Sparactus with modern knowledge to recommend C++ books as an experienced warrior. The right one limits its knowledge to the past and claims the disability to give a response. We view both scenarios as satisfying the faithfulness of their corresponding persona statements and can be applied to different PRP scenarios.

## 7   Conclusion

This paper proposes a pioneering study on quantifying and optimizing the global faithfulness of PRP methods. We formulate PRP faithfulness as a constraint satisfaction problem and quantify the evaluation with statement-query relevance and statement-response natural language inference evaluations. Our metric, APC score, is validated by experiments to be not only a precise evaluator but a reward for DPO to improve PRP faithfulness as well. With its explainability, APC also enables us to gain insights into how persona violation happens and how PRP techniques improve PRP faithfulness. Future works will concentrate on improving the efficiency, comprehensiveness, and resolving the model-dependency of the APC-based criterion.

## Acknowledgement

This work aims to contribute not only to the research community but also to a broader ACG community by introducing more powerful role-playing agents. It is also done in memory of the 16th *Koishi's Day* (May 14th), 2024, since the release of TH11, Touhou Chireiden ∼ Subterranean Animism[4] in 2008.

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

# A    Limitation and Future Work

While our APC criterion is a fine-grained and explainable evaluation for PRP faithfulness, several limitations are still awaiting refinement in future works.

**Efficiency**    The strict APC score in our experiments has to be assigned by traversing through all persona statements to assign the relevance and NLI scores. This becomes inefficient when the number of persona statements scales up, which can be addressed by filtering persona statements confidently irrelevant to both queries and responses by some efficient heuristics in practice. Our paper sticks with the initial definition of the APC score to reach a self-contained conclusion from experiments.

**Simplification**    The summing up of satisfaction probability to persona statement might be a simplification as different persona statements might have different importance for the response. Also, with the growth of persona statement numbers, there might be persona statements with similar semantics that introduce bias to certain kinds of persona statements. Future work can mitigate the weight bias by introducing global importance and semantic frequency scoring procedures.

**Model-dependent Evaluation**    While our PRP methods are evaluator-agnostic, some models are distilled from GPT-4, which is also used to build the discriminators for evaluation. While GPT-4 has shown high alignment with humans, our evaluation might still introduce the preference from GPT-4's view, which is a shared limitation of LLM-based evaluation.

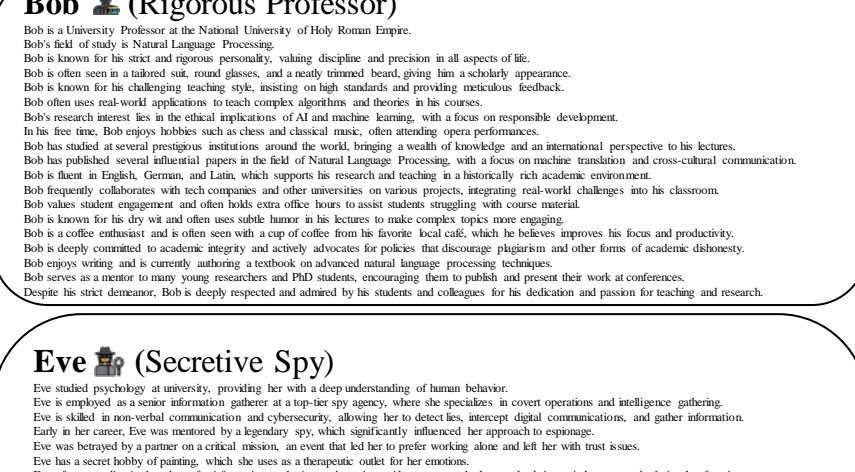

**Alice** 🧑‍🎤 (Introverted Guitarist)

Alice Carter is a 26-year-old professional guitarist who grew up in a small coastal town.
Alice has long, wavy dark brown hair, deep green eyes, and typically wears comfortable, loose clothing like maxi dresses or jeans with vintage band tees.
Alice is introverted, reflective, sensitive, and artistic, often using her music to express emotions she finds difficult to communicate verbally.
Alice performs occasionally at local venues and teaches guitar lessons, preferring intimate settings over large concerts.
Besides playing guitar, Alice's hobbies include reading, particularly poetry and classic literature, and sketching in her notebooks.
Alice's music is a blend of acoustic indie and folk, influenced by her coastal upbringing and introspective nature, with introspective and poetic lyrics.
Alice has a close group of friends who share her passion for music and art, and despite her introverted nature, she values these relationships deeply.
Alice aspires to record an album that captures her experiences and emotions, hoping her music will resonate with others who feel misunderstood or on the fringes.

**Bob** 👨‍🏫 (Rigorous Professor)

Bob is a University Professor at the National University of Holy Roman Empire.
Bob's field of study is Natural Language Processing.
Bob is known for his strict and rigorous personality, valuing discipline and precision in all aspects of life.
Bob is often seen in a tailored suit, round glasses, and a neatly trimmed beard, giving him a scholarly appearance.
Bob is known for his challenging teaching style, insisting on high standards and providing meticulous feedback.
Bob often uses real-world applications to teach complex algorithms and theories in his courses.
Bob's research interest lies in the ethical implications of AI and machine learning, with a focus on responsible development.
In his free time, Bob enjoys hobbies such as chess and classical music, often attending opera performances.
Bob has studied at several prestigious institutions around the world, bringing a wealth of knowledge and an international perspective to his lectures.
Bob has published several influential papers in the field of Natural Language Processing, with a focus on machine translation and cross-cultural communication.
Bob is fluent in English, German, and Latin, which supports his research and teaching in a historically rich academic environment.
Bob frequently collaborates with tech companies and other universities on various projects, integrating real-world challenges into his classroom.
Bob values student engagement and often holds extra office hours to assist students struggling with course material.
Bob is known for his dry wit and often uses subtle humor in his lectures to make complex topics more engaging.
Bob is a coffee enthusiast and is often seen with a cup of coffee from his favorite local café, which he believes improves his focus and productivity.
Bob is deeply committed to academic integrity and actively advocates for policies that discourage plagiarism and other forms of academic dishonesty.
Bob enjoys writing and is currently authoring a textbook on advanced natural language processing techniques.
Bob serves as a mentor to many young researchers and PhD students, encouraging them to publish and present their work at conferences.
Despite his strict demeanor, Bob is deeply respected and admired by his students and colleagues for his dedication and passion for teaching and research.

**Eve** 🕵️ (Secretive Spy)

Eve studied psychology at university, providing her with a deep understanding of human behavior.
Eve is employed as a senior information gatherer at a top-tier spy agency, where she specializes in covert operations and intelligence gathering.
Eve is skilled in non-verbal communication and cybersecurity, allowing her to detect lies, intercept digital communications, and gather information.
Early in her career, Eve was mentored by a legendary spy, which significantly influenced her approach to espionage.
Eve was betrayed by a partner on a critical mission, an event that led her to prefer working alone and left her with trust issues.
Eve has a secret hobby of painting, which she uses as a therapeutic outlet for her emotions.
Eve often uses disguised gadgets for information gathering and evasion, with a smartwatch that can hack into wireless networks being her favorite.
To the public, Eve presents herself as a successful cybersecurity consultant, a persona that helps her gather intelligence and conceal her true identity.
Eve experienced the loss of a sibling in a spy incident, which fuels her pursuit of justice and influences her risk-taking approach in operations.
Eve's ultimate goal is to dismantle a global crime syndicate that has long evaded the agency, a mission that is both professional and deeply personal to her.
Eve is fluent in several languages, including Russian, Mandarin, and Arabic, which aids her in blending into different cultures during her fieldwork.
Eve has contributed to the development of new spy technologies, such as a micro-drone for surveillance that is no larger than a butterfly.
Eve is trained in Krav Maga and Brazilian Jiu-Jitsu, enabling her to defend herself in close-quarters combat.
Eve often contemplates the ethical implications of her work and reads extensively on philosophy and ethics.
Eve maintains a complex relationship with her family, who are unaware of her true occupation, and she keeps her distance to ensure their safety.
Eve has a history of complicated romantic relationships, primarily with other spies or operatives, which have sometimes affected her professional and personal life.
Eve is a history enthusiast, particularly of the Cold War era, and she uses historical knowledge to plan and execute her missions.
Inside her agency, Eve has secretly befriended a few colleagues with whom she shares similar doubts and fears.
Eve is a master of disguise, capable of drastically altering her appearance to the point where even close acquaintances struggle to recognize her.
Beyond her immediate spy duties, Eve aspires to start a private security firm to protect global human rights activists from espionage and assassination.
Eve is an excellent cook who specializes in dishes from the regions she infiltrates.
Eve plays the violin, a skill she uses as a cover identity during her missions.
Eve has received advanced training in psychological operations, which she uses to manipulate targets and extract information.
Eve has a significant fear of water due to a near-drowning incident in her early childhood.
Eve is an avid reader and often uses literary quotes to code her messages.
Eve is passionate about wildlife conservation and aligns her missions with environmental objectives when possible.
Eve has trained her memory to perform at peak levels, using techniques like the method of loci for her missions.
As a child, Eve dreamed of becoming an astronaut, a desire that is fulfilled in a different way through her career in espionage.
Eve collects artifacts from her missions, viewing them as souvenirs and lessons in history and human behavior.
Eve mentors younger agents, teaching them both the skills and ethical complexities of their job..

Figure 7: The persona statements of original characters.

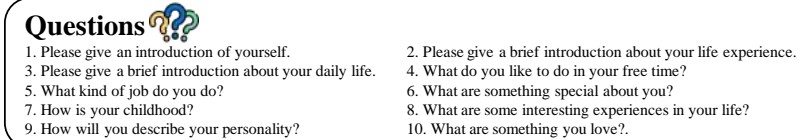

**Questions** 🧐❓

1. Please give an introduction of yourself.
2. Please give a brief introduction about your life experience.
3. Please give a brief introduction about your daily life.
4. What do you like to do in your free time?
5. What kind of job do you do?
6. What are something special about you?
7. How is your childhood?
8. What are some interesting experiences in your life?
9. How will you describe your personality?
10. What are something you love?.

Figure 8: The interview questions for original characters.

## B Original Characters and Interview Queries

The persona statements and interview questions for original characters are presented in Figures 7 and 8. We brainstorm the persona statements and prompt GPT-4 only to formalize them as natural language. As the original characters have few persona statements, we propose the 10 most important questions to evaluate PRP faithfulness. The information about famous figures in our experiments can be found in (Shao et al., 2023).

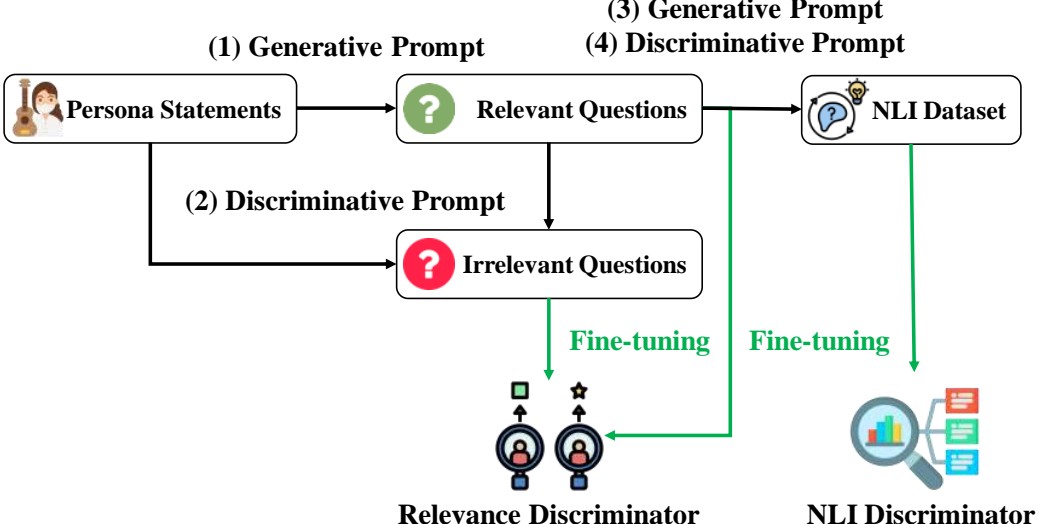

Figure 9: The symbolic distillation pipeline to build discriminators.

## C  Symbolic Distillation

We apply prompts in the Appendix H for symbolic distillation from GPT-4 to build statement-query relevance and statement-to-response NLI discriminators. The whole pipeline includes 4 stages.

- **Generative Prompt for Relevance Dataset** We prompt GPT-4 to generate 3 questions relevant to each persona statement.
- **Discriminative Prompt for Relevance Dataset** For each generated query, we randomly select 5 other persona statement and prompt GPT-4 to discriminate the query as relevant or irrelevant. Most statement-query pairs are discriminated as irrelevant in this stage.
- **Generative Prompt for NLI Dataset** Based on each relevant statement-query pair, we prompt GPT-4 to generative responses entailed, neutral, and contradicted by the persona statement.
- **Discriminative Prompt for NLI Dataset** For each query-response pair, we randomly select 3 other persona statements and prompt GPT-4 to discriminate the response as entailed, neutral or contradicted. Most statement-to-response pairs are discriminated as neutral in this stage.

These datasets, with statistics shown in Appendix F, are then used to fine-tune the discriminators. For the evaluation, the seed persona statements are based on three characters: Beethoven, Newton, and Socrates. For each character used to learn PRP methods, the datasets are prompted based on only the persona statements of that character. The RAG retriever is fine-tuned on the statement-query relevance dataset. For APC-based DPO, the discriminators are built in the same way as the evaluator. The hyperparameters are presented in Appendix D.

| Character | Alice | Bob | Eve | Beethoven | Newton | Socrates |
|---|---|---|---|---|---|---|
| #Persona Statement | 8 | 19 | 30 | 383 | 354 | 324 |
| #Question | 10 | 10 | 10 | 77 | 90 | 89 |
| #Relevance Data | 64 | 152 | 240 | 3061 | 2832 | 2591 |
| #NLI Data | 144 | 459 | 545 | 6774 | 6331 | 5760 |

| Character | Spartacus | Hermione | Voldemort | Cleopatra | Caesar | MLK |
|---|---|---|---|---|---|---|
| #Persona Statement | 77 | 146 | 201 | 374 | 498 | 599 |
| #Question | 89 | 118 | 77 | 93 | 87 | 92 |
| #Relevance Data | 616 | 1167 | 1608 | 2991 | 3981 | 4789 |
| #NLI Data | 1368 | 2586 | 3546 | 6660 | 8856 | 10644 |

Table 3: The statistics of characters in our experiments.

## D  More PRP Method Implementation Details

**Fine-tuning Gemma**    is applied for PRP models (EU and DPO). Different fine-tuning procedures for Gemma share the same set of hyperparameters. 128-rank LoRA is used to fine-tune the model with AdamW (Loshchilov & Hutter, 2019) as the optimizer, learning rate initialized as $2 \times 10^{-4}$. Based on the number of persona statements, EU for original characters fine-tunes for 20 epochs, while for famous figures fine-tunes for 5 epochs. DPO fine-tunes for 10 epochs for all characters.

**Fine-tuning DeBERTa**    is applied for discriminators and RAG retrievers. Different fine-tuning procedures for DeBERTa also share the same set of hyperparameters. The DeBERTa discriminators are fully fine-tuned with AdamW as the optimizer, learning rate initialized as $1 \times 10^{-5}$. The statement-query relevance discriminator is fine-tuned for 5 epochs and the statement-to-response NLI discriminator is fine-tuned for 10 epochs.

**Preference Assignment**    We sample two responses from a PRP agent with temperature 1.0, the sample with a higher APC score is assigned as the preferred one when the difference is larger than a threshold for filtering, which is set to 0.2 in our implementation. We build 100 preference pairs before the filtering for APC-based DPO.

## E  Human Evaluation

The human evaluation is applied only to the simple original characters because memorizing all their persona statements and applying them to evaluating famous figures are too challenging for humans. For each response, the response is scored following the scheme,

- **Score: 0 (Wrong Character)** The response completely represents another character (including LLM), or is not role-playing as any character.
- **Score: 2 (Incorrect Information)** The response is role-playing as the character, but the information included is completely incorrect.
- **Score: 4 (Hallucinated Information)** The response is role-playing as the character, but the information included is partially incorrect.
- **Score: 6 (Hallucinated Details)** The response is role-playing as the character, but a few details are incorrect, or some important information is missed.
- **Score: 8 (Trustful Information)** The response is role-playing as the character with all the information mentioned is correct but a few details are missed.
- **Score: 10 (Completely Faithful)** The response is role-playing as the character with all important information is mentioned faithfully.

The score is averaged over responses as the final human evaluation metric.

## F  Statisitcs of Characters in Experiments

We present the statistics of the characters in our experiments in Table 3

| Character | Alice | | | Bob | | | Eve | | |
| #Statement | 8 | | | 19 | | | 30 | | |
| Evaluator | APC | | Human | APC | | Human | APC | | Human |
| | DeB | GPT-4 | | DeB | GPT-4 | | DeB | GPT-4 | |
| **w/o CPO** Gemma-7B | 4.3 | 3.1 | 1.8 | 9.7 | 7.3 | 1.8 | 14.2 | 10.6 | 2.0 |
| EU | 6.2 | 3.9 | 6.4 | 12.0 | 8.0 | 6.2 | 17.1 | 11.5 | 4.6 |
| LCM | 6.2 | 4.2 | 6.8 | 13.1 | 9.1 | 7.2 | 17.4 | 11.5 | 5.0 |
| RAG | 6.4 | 4.6 | 6.8 | 12.6 | 8.6 | 6.8 | 18.3 | 13.2 | 5.8 |
| **w/ CPO** EU | 6.3 | 4.2 | 6.8 | 12.4 | 8.7 | 6.8 | 17.4 | 11.7 | 5.2 |
| | (+0.1) | (+0.3) | (+0.4) | (+0.4) | (+0.7) | (+0.6) | (+0.3) | (+0.2) | (+0.6) |
| LCM | 6.4 | **5.0** | **7.6** | **13.9** | 9.4 | 7.8 | 18.6 | 14.1 | 6.6 |
| | (+0.2) | (+0.8) | (+0.8) | (+0.8) | (+0.3) | (+0.6) | (+1.2) | (+2.6) | (+1.6) |
| RAG | **6.5** | **5.0** | **7.6** | 13.8 | **10.7** | **8.2** | **19.3** | **15.0** | **7.0** |
| | (+0.1) | (+0.4) | (+0.8) | (+1.2) | (+2.1) | (+1.2) | (+1.0) | (+1.8) | (+1.2) |

Table 4: PRP Faithfulness Evaluation with the full APC score on simple and contamination-free characters.

| Character | Spartacus | Hermione | Voldemort | Cleopatra | Caesar | MLK | Average |
| #Statement | 77 | 146 | 201 | 374 | 498 | 599 | |
| **GPT** ChatGPT | 69.1 | 128.4 | 168.6 | 324.1 | 421.7 | 473.1 | 264.2 |
| GPT-4 | 69.0 | 129.5 | 169.6 | 326.2 | 425.1 | 476.3 | 266.0 |
| **w/o CPO** Gemma-7B | 68.8 | 129.3 | 173.0 | 327.1 | 423.5 | 470.8 | 265.4 |
| EU | 67.3 | 125.9 | 166.1 | 321.5 | 418.4 | 468.0 | 261.2 |
| RAG | **70.1** | 130.0 | 174.6 | **328.1** | 425.4 | 477.5 | 267.6 |
| **w/ CPO** Gemma-7B | 69.4 | 130.2 | 176.4 | 326.7 | 423.1 | 479.3 | 267.5 |
| EU | 68.7 | 127.8 | 170.9 | 324.5 | 418.7 | 468.1 | 263.1 |
| RAG | 69.9 | **130.9** | **176.6** | 327.7 | **426.4** | **481.1** | **268.8** |

Table 5: PRP Faithfulness Evaluation with the full APC score on characters with persona statements at scale.

# G   Full Award Result

In Tables 4 and 5, we report the full APC scores gained by different PRP methods. We observe the proportion of satisfied constraints is negatively correlated with the number of persona statements. This indicates PRP becomes more difficult with the growth of persona statement numbers. Also, original characters are harder to be faithfully role-played than those memorized characters, which indicates the significant influence of LLM memorization on PRP.

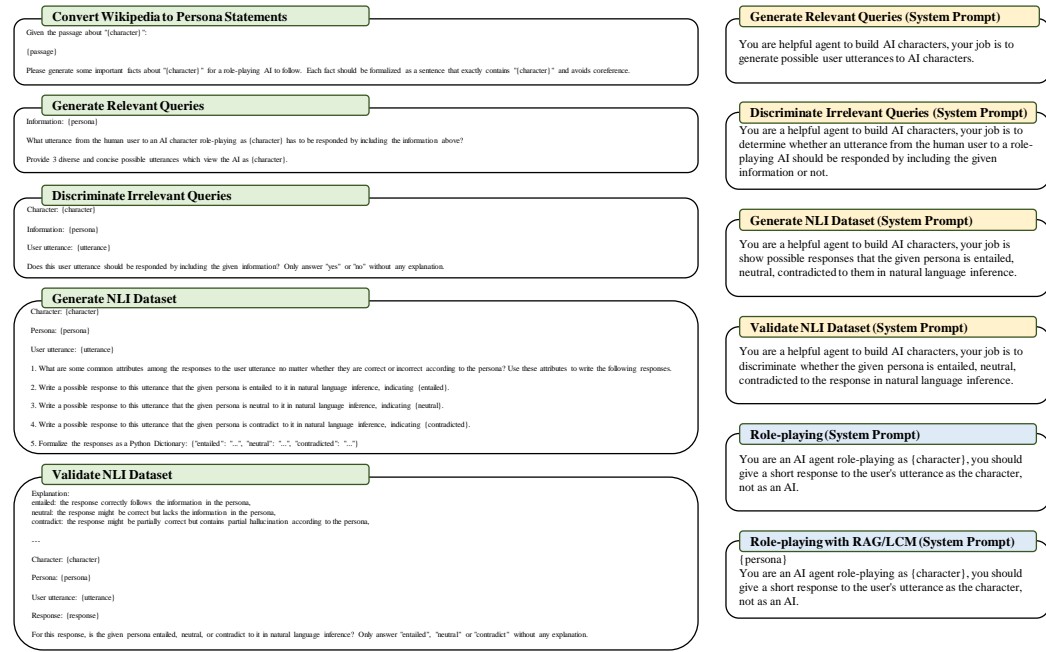

Figure 10: The prompts used in our experiments.

# H Prompts

The prompts in our experiments are shown in Figure 10. The prompts include the generative or discriminative goals, and also the formalization procedure for decoding into JSON files.

# I More Characters

Besides characters in the main content, we further expand the scope of characters to different ethnicity, which involves,

- Alex: An African American baseball player
- Isabella: An Italian traveling cook
- Takayoshi: A Japanese game developer
- Ousmane: A rich gold mine owner of the Malian Empire in the 1300s
- Jones: A young British worker in the Victorian Era
- Zhe: A Chinese poet in the Tang Dynasty
- Crossan: A time-traveling scientist
- Betty: A pet cat who can talk with ghosts
- X: An alien space traveler and photographer

These characters can better represent people with different spatial and temporal backgrounds and even cover non-human characters from the fantasy world.

| Character | | Alex | Isabella | Takayoshi | Ousmane | Jones | Zhe | Crossan | Betty | X |
|---|---|---|---|---|---|---|---|---|---|---|
| w/o CPO | Vanilla | 0.5 | 0.8 | 0.6 | 0.3 | 0.9 | 1.1 | 0.3 | 0.3 | 0.7 |
| | EU | 1.8 | 2.8 | 2.0 | 1.4 | 0.7 | 3.8 | 2.0 | 1.2 | 5.2 |
| | LCM | 7.1 | 7.4 | 6.5 | 4.5 | 6.2 | 5.2 | 2.2 | 2.8 | 8.1 |
| | RAG | 7.6 | 8.1 | 6.9 | 3.0 | 6.6 | 5.8 | 1.8 | 3.2 | 7.5 |
| w/ CPO | EU | 5.3 | 6.1 | 5.7 | 3.6 | 4.8 | 4.9 | 3.1 | 2.9 | 7.9 |
| | LCM | 7.5 | 7.7 | 7.0 | 4.8 | 6.2 | 5.4 | 4.5 | 3.9 | 8.2 |
| | RAG | 7.9 | 8.2 | 7.4 | 3.9 | 7.5 | 6.9 | 2.5 | 4.6 | 8.9 |

Table 6: PRP performance on more characters based on the distilled DeBERTa Evaluator

| Character | | Alex | Isabella | Takayoshi | Ousmane | Jones | Zhe | Crossan | Betty | X |
|---|---|---|---|---|---|---|---|---|---|---|
| w/o CPO | Vanilla | 0.2 | 0.1 | -0.2 | 0.5 | 0.2 | 0.2 | 0.4 | 0.1 | 0.8 |
| | EU | 1.4 | 1.8 | 3.0 | 0.5 | 1.3 | 6.4 | 1.2 | 0.3 | 7.4 |
| | LCM | 3.1 | 8.6 | 5.6 | 4.1 | 7.4 | 3.4 | 2.1 | 1.6 | 11.3 |
| | RAG | 3.3 | 7.8 | 6.1 | 1.6 | 8.1 | 4.3 | 2.7 | 2.2 | 10.1 |
| w/ CPO | EU | 2.7 | 5.6 | 5.9 | 3.0 | 4.7 | 7.1 | 2.2 | 1.5 | 9.5 |
| | LCM | 3.2 | 9.8 | 8.1 | 4.6 | 8.2 | 7.8 | 4.0 | 2.3 | 12.1 |
| | RAG | 4.8 | 10.0 | 9.8 | 2.0 | 8.3 | 7.3 | 2.9 | 3.1 | 14.6 |

Table 7: PRP performance on more characters based on the GPT-4 Evaluator

The experiment results are presented in Tables 6 and 7, which is consistent with our results in Tables 1 and 2. Thus, our conclusion is certificated on a larger scope for broader application.

## J  Metric Comparison

To better justify selecting our APC score and also support the claim that the fine-grained APC score has the advantage over coarse-grained metrics, we add a coarse-grained metric as the baseline. We directly prompt GPT-4 with the criterion used for human evaluation shown in Appendix E. We also distill this scoring ability (following the same scenario as APC) to DeBERTa to check whether the efficiency can be boosted. We evaluate the Spearman correlation between the metric and the human evaluation of the 7 role-playing methods on the 3 human-evaluated characters.

| Character (#Persona Statement) | | Alice (8) | Bob (19) | Eve (30) |
|---|---|---|---|---|
| GPT-4 | Coarse-grained Score | 92.42 | 86.27 | 81.40 |
| | APC Score | 97.18 | 99.10 | 99.10 |
| DeBERTa | Coarse-grained Score | 81.40 | 69.91 | 54.57 |
| | APC Score | 88.61 | 95.50 | 99.10 |

Table 8: Comparison of PRP metrics on the consistency with human evaluation.

The results verify that 1) Fine-grained APC score shows better consistency with human evaluation. 2) The fine-grained APC score is stable to the number of persona statements while the coarse-grained score degrades with the increase of persona statements. 3) The coarse-grained evaluating ability is harder to be distilled into smaller models for efficiency boosting. Based on case checking, we find an underlying issue of the coarse-grained metric is the LLM will assign a high score to a response once it contains some correct information, ignoring the missing important information (active constraint) and occasionally conflictions (passive constraint).

# K    Student Model Comparison

We select DeBERTa as the student model to distill from GPT-4 because small encoders (BERT, RoBERTa, etc.) show promising performance on relevance and NLI, which are classic NLU tasks in the GLUE benchmark. Among encoders, DeBERTa (DeBERTa-v3-large) is a state-of-the-art model that shows strong performance after fine-tuned on NLU tasks. To further verify DeBERTa as a proficient student model, we add an analysis of the in-domain (ID)/out-of-domain (OOD) performance and the efficiency of different base models for distillation.

| Task | Relevance | | NLI | | Efficiency |
|------|-------|-------|-------|-------|------------|
|      | ID | OOD | ID | OOD | |
| **DeBERTa (Base)** | 92.46 | 89.90 | 89.72 | 87.80 | 409.6it/s |
| **DeBERTa (Large)** | 94.04 | 92.10 | 93.46 | 91.50 | 150.8it/s |
| **Gemma-1.1-it (2b)** | 94.25 | 92.50 | 93.68 | 91.80 | 26.4it/s |

Table 9: Model Performance Comparison

The in-domain test set (1697 instances for Relevance, 3773 instances for NLI) is the 20% split of the characters (Beethoven, Newton, Socrates) that build the training set (6787 instances for Relevance, 15092 instances for NLI). The out-of-domain test set samples 1000 cases from other characters. The results show DeBERTa-V3-Large (300M) shows a comparative performance with a 2B Gemma model, while is about 6 times faster, which justifies DeBERTa to be a strong student model. The out-of-domain performance is generally high, which indicates the generalizability to other characters. Finally, an extra discovery is that DeBERTa-v3-base (100M) can further significantly boost efficiency with some trade-offs in accuracy.

