# OpenReview forum: "Quantifying and Optimizing Global Faithfulness in Persona-driven Role-playing"
_NeurIPS.cc/2024/Conference — NeurIPS 2024 poster_

### Official Review · Reviewer_Pxme · 2024-07-09

**Soundness:** 3
**Presentation:** 3
**Contribution:** 3
**Rating:** 6
**Confidence:** 4

**Summary:**

This paper introduces the Active-Passive-Constraint (APC) score to evaluate and optimize the faithfulness of AI-driven persona interactions in role-playing applications. The authors propose a novel method by quantifying interactions using a fine-grained, constraint-based scoring system, significantly advancing the granularity of PRP evaluations. The methodology employs a NLI model distilled from GPT-4 for efficient and consistent evaluations, validated through experiments that demonstrate a high correlation with human judgment.
Another key contribution is using  APC score as a preference target for Direct Preference Optimization (DPO)， offering a new insight into improving PRP systems.

**Strengths:**

1. Assessing role-playing is a highly important yet challenging task, and the method proposed in this paper is simple, user-friendly, and quantifiable. This contributes significantly to the rapid iteration within the Role Playing LLM field.
2. This study is not limited to an assessment method; it also serves as an optimization target that improves the global faithfulness of AI characters.

**Weaknesses:**

1. The most concern is that the main experiment covers too few characters(3 simple characters and 6 famous characters, total 9), which makes it difficult to ensure the reliability of the conclusions.

2.The APC Score relies on an additionally trained NLI Discriminator, which, although claimed to have an accuracy of 90% in this paper, lacks provided details on its training and the volume of data used. There is a question regarding whether this 300M classifier model can generalize to a broader range of characters.

3.The paper lacks citation of the research[1] that is similar in core idea to this study, which also focuses on enhancing role-playing effectiveness by limiting the knowledge boundaries of characters to ensure their responses are confined to what they are known.

[1]Lu K, Yu B, Zhou C, et al. Large language models are superpositions of all characters: Attaining arbitrary role-play via self-alignment[J]. arXiv preprint arXiv:2401.12474, 2024.

**Questions:**

1. Why is Regularization APC considered necessary, and how should it be chosen in relation to the APC score when in use?

2. APC-Based DPO is one of the primary contributions of this study; however, Appendix D mentions that only 100 preference data points were used for training. Could this small amount of data lead to significant model overfitting?

3. Table 1, 4, and 5 show a huge difference in APC scores between simple characters and famous characters (6 vs 400). What could be the possible reasons for this discrepancy, and what does the absolute magnitude of the APC score signify?

**Limitations:**

The author elaborates the limilation of this study.

---

> ### Author Rebuttal · Authors · 2024-08-06
>
> We are very pleased to receive your positive recommendation for our work. Your suggestions and questions are insightful and valuable to further improve the quality of our paper's content and writing. We will address your concerns with the following clarification and experiments.
>
> ## **Character Coverage**
>
> We broaden the character coverage in our experiments by adding 9 new original characters, as in the **Experiment Comprehensiveness** section of **Author Rebuttal by Authors**. We not only concentrate on the numbers of the characters but also design the characters to be with different spatial (Asian, European, African, etc.) and temporal (past, present, future) conditions. We also include fantasy characters like a time-traveling scientist, thus broadening the generalizability verified from the experiment.
>
> The results of these new diverse characters **(1st and 2nd tables in Author Rebuttal by Authors)** achieve a consistent conclusion with the existing experiments, thus further strengthening the generalizability of our discovery with a broader scope of characters. We will further include human evaluation for these new characters in the final version of our paper.
>
> ## **Generalizability of the distilled discriminators**
>
> For the student discriminator distilled to perform relevance and NLI evaluation, we choose DeBERTa because it generalizes well on classic NLU tasks including Relevance and NLI, where pre-trained encoders like BERT and RoBERTa show strong generalizability (out-of-domain MNLI for example). To further justify the generalizability of DeBERTa in our experiments, we included a comparison of the distillation efficiency and evaluation efficiency of student models in the **Implementation Justification** section (including the detailed datasets statistics) of the **Author Rebuttal by Authors**. The fine-tuning hyperparameters for distillation can be found in current Appendix D.
>
> The training results in the **4th table of the Author Rebuttal by Authors** show that DeBERTa can rival LLMs like Gemma in cross-character performance while greatly improving evaluation efficiency. Therefore, we selected DeBERTa as the student model for evaluation efficiency.
>
> ## **Missing citations**
>
> Thank you for mentioning the missing citations, we will cite more papers that enhance role-playing effectiveness by limiting the knowledge boundaries, which include [1].
>
> ## **Questions**
>
> **Why is regularized APC considered necessary?** Regularized APC ($\Delta$APC) represents the difference between the method's APC score and the APC score of an oracle that always gives a neutral response in NLI. Thus APC and $\Delta$APC have no difference when ranking role-playing methods. As mentioned in section 4.2 of our paper, $\Delta$APC is introduced to improve the meaning of the absolute value (How the method is better than a dumb system that always outputs "Hello"?). As shown in Table 1 in the current paper, the vanilla Gemma model without any knowledge about the original characters shows ~0 $\Delta$APC score. For evaluation, $\Delta$APC is generally preferred because it shows the difference with the simplest baseline. There are also cases preferring the unregularized APC score like estimating the difference with a perfect role-playing system which gains an unregularized APC score equal to the number of persona statements.
>
> **Could the small amount of DPO data lead to significant model overfitting?** Even though 100 training cases may seem small, each case in DPO covers a different question relevant to multiple persona statements, making it more comprehensive than it appears. Also, DPO does not continuously amplify the possibility of the preferred cases but learns a relative difference between the preferred and the rejected cases, which avoids overfitting. In our experiments, the questions used for DPO annotation and for testing are different, and the DPO still benefits the testing result. This experiment result also indicates the DPO is not overfitting but generalizes the global faithfulness to unseen questions. Finally, as the DPO data are easy to collect (only requires a question to prompt the role-playing model), one can easily increase the number of data for DPO to further generalize global faithfulness.
>
> **Why do Tables 1, 4, and 5 show a huge difference in APC scores?** Table 1 shows a regularized APC score, which is the difference between the method's APC score and the APC score of an oracle that always gives a neutral response in NLI. Tables 4 and 5 show the unregularized APC score and thus is higher. The huge difference in APC scores between simple characters and famous characters is a result of the value scaling of the APC score with the number of persona statements. APC score essentially estimates the number of persona statements satisfied by the response, thus those famous characters with more persona statements will generally have higher unregularized APC scores.

---

> > ### Comment · Reviewer_Pxme · 2024-08-12
> >
> > I'm curious why it's difficult to scale up the experiment. What is the most cost part?  or just there is no suitable dataset.

---

> > > ### Author Response · Authors · 2024-08-12
> > > **Response to Reviewer Pxme**
> > >
> > > Thanks for your question! The main difficulty to scale-up the experiments is the cost to fine-tune character LLMs. Take experience uploading (EU) as an example, for each character, we need to train it with separate synthesized experiences.
> > >
> > > Also, the lack of well-curated dataset is another problem, we follow the influential character-LLM work [1] to take the involved characters (9 characters) with extra original characters into our experiments. This issue can be addressed with the emergence of new well-curated datasets.
> > >
> > > [1] Character-LLM: A Trainable Agent for Role-Playing

---

> > > > ### Comment · Reviewer_Pxme · 2024-08-13
> > > >
> > > > Thanks for your response, I would keep my positive rating.

---

### Official Review · Reviewer_iNA2 · 2024-07-10

**Soundness:** 3
**Presentation:** 2
**Contribution:** 3
**Rating:** 7
**Confidence:** 3

**Summary:**

The paper introduces an evaluation method for persona-driven role-playing (PRP) using the Active-Passive-Constraint (APC) scoring system. This system measures the faithfulness of AI responses to predefined persona statements by calculating APC scores and applying Direct Preference Optimization (DPO) to improve AI character adherence to personas. The authors validate the effectiveness of the APC scoring system through various experiments, demonstrating its applicability and improvements over existing methods.

**Strengths:**

The proposed APC scoring system offers a nuanced approach to evaluating PRP, addressing limitations in existing coarse-grained methods by providing a detailed and explainable metric.

The application of APC-based DPO as a reward system for enhancing AI character faithfulness is innovative and effectively demonstrated through experiments.

The paper introduces Contextual Preference Optimization, further refining the evaluation and optimization process, showcasing a comprehensive approach to improving PRP methods.

The detailed methodology for calculating APC scores and integrating them with DPO is well-structured and validated, providing a robust framework for future research.

**Weaknesses:**

The complexity of the proposed methodology might limit its accessibility and reproducibility. Simplifying or providing clearer explanations for key components could enhance understanding and adoption.

The paper lacks justification for selecting specific models like Deberta and Gemma. A comparison with other potential models could strengthen the argument for their use.

The experiments presented are minimal, making it difficult to generalize the findings. More extensive experiments, including diverse scenarios and models, would provide stronger support for the proposed methodology.

There are redundant explanations regarding Long-Context Memory (LCM) and Retrieval-Augmented Generation (RAG) methods. Streamlining these sections could improve the paper's clarity and focus.

The results in the tables are not sufficiently explained. Better captions, detailed discussions, and visual aids could enhance the readability and interpretability of the data, ensuring that each value's significance is clear.

The paper does not provide a clear comparison with simpler baseline methods for evaluating PRP faithfulness. Including such comparisons would highlight the improvements and advantages of the APC scoring system.

**Questions:**

N/A

---

> ### Author Rebuttal · Authors · 2024-08-06
>
> We are grateful for your positive attitude towards the quality and contribution of our work. We also find your suggestions are insightful and beneficial to polish our work. We will include the following experiments and clarification to further solidify our conclusions and address your concerns.
>
> ## **Experiment scope**
>
> We add 9 new original characters to our experiments, as in the **Experiment Comprehensiveness** section of **Author Rebuttal by Authors**. We not only concentrate on the numbers of the characters but also design the characters to be with different spatial (Asian, European, African, etc.) and temporal (past, present, future) conditions. We also include fantasy characters like a time-traveling scientist, thus broadening the generalizability verified from the experiment.
>
> The results of these new diverse characters **(1st and 2nd tables in Author Rebuttal by Authors)** achieve a consistent conclusion with the existing experiments, thus further strengthening the generalizability of our discovery with a broader scope of characters. We will further include human evaluation for these new characters in the final version of our paper.
>
> ## **Metric comparison**
>
> We follow your suggestion to add a simple baseline - directly prompting the LLM to produce a coarse-grained score as the baseline in the **Metric Justification** section in **Author Rebuttal by Authors**, whose limitation is discussed in the current introduction section.
>
> As the results presented in the **3rd table in Author Rebuttal by Authors**, the fine-grained APC score is more consistent with human evaluation and stable to the number of persona statements. Thus, our claim that a fine-grained metric like the APC score will be better for role-playing faithfulness evaluation is better supported.
>
> ## **Justification for selecting specific models**
>
> For the student model, we choose DeBERTa because it empirically performs well on classic NLU tasks like Relevance and NLI, where pre-trained encoders like BERT and RoBERTa excel (according to the GLUE benchmark). To further explain why we used DeBERTa in our experiments, we included a comparison of the distillation efficiency and evaluation efficiency of student models in the **Implementation Justification** section of the **Author Rebuttal by Authors**.
>
> The results in the **4th table of the Author Rebuttal by Authors** show that DeBERTa can match LLMs like Gemma in distillation performance while greatly improving evaluation efficiency. Therefore, we selected DeBERTa as the student model for efficient experiments.
>
> For the role-playing base model, the requirement to support our claims is the knowledge of the famous figures and the ignorance of the original characters. Thus, most existing LLMs meet this requirement and we select Gemma, which is a state-of-the-art LLM at the time of the experiments. In the final version of our paper, we will include experiments on recent LLMs that show strong abilities in various domains, such as Llama-3 and Mistral, to further verify the generalizability of our conclusion.
>
> ## **Writing and content organization**
>
> We will simplify the flow to introduce the key components to make it easier to understand our contributions. More specifically, we will include a table for denotations which the readers can refer to to understand the attributes and function of the introduced key components. We will also reduce the redundancy when explaining RAG and LCM by streamlining these sections, such as referring to section 3.2 in section 5.2 and concentrating more on the method setup. Finally, we will further polish the quality of captions, detailed discussions, and visualization. For example, we add more details to the caption of Figure 3 to further clarify the explanations of the two subfigures.

---

> ### Comment · Reviewer_iNA2 · 2024-08-12
>
> Thank you for considering the suggestions and explaining in detail about the new additions in detail, after considering the new updates, I am increasing my score to 7

---

### Official Review · Reviewer_Ehwr · 2024-07-12

**Soundness:** 3
**Presentation:** 3
**Contribution:** 3
**Rating:** 7
**Confidence:** 4

**Summary:**

The paper presents a novel approach to evaluating and optimizing the faithfulness of persona-driven role-playing (PRP) in AI characters. It addresses the limitations of existing coarse-grained faithfulness criteria. The authors introduce the Active-Passive-Constraint (APC) score, which discriminates persona statements into active and passive constraints based on their relevance to user queries. The paper validates the APC score through experiments, demonstrating high correlation with human evaluation and consistency with GPT-4's discrimination. It further leverages the APC score in direct preference optimization (DPO) to enhance AI character responses, revealing DPO as a competitive technique for adhering to constraints and complementing other methods.

**Strengths:**

1. It introduces a pioneering Active-Passive-Constraint (APC) score, providing a fine-grained and quantifiable measure of PRP faithfulness.
2. The paper successfully demonstrates the APC score's alignment with human judgment through rigorous experiments and its practical utility in optimizing AI behavior through direct preference optimization (DPO).
3. The comprehensive analysis, case studies, and the paper's ability to reveal the advantages and limitations of existing PRP techniques further underscore its strengths.

**Weaknesses:**

1. The APC score's simplicity in aggregating satisfaction probabilities might not fully capture the varying importance of different persona statements to the response.
2. Additionally, the model-dependent nature of the evaluation, relying on GPT-4 for discriminators, could introduce biases aligned with GPT-4's training.
3. Except for persona consistency, other evaluation dimensions such as response activeness, context consistency, are also important to the experience of PRP.

**Questions:**

1. why not choose a larger model to compute the APC score? It may obtain higher consistency with GPT-4 or human?
2. why not train the model of computing APC score by using the label annotated by human?
3. Have you considered a more comprehensive evaluation score for role-playing dialogue, not just in the persona dimension.

**Limitations:**

The authors have acknowledged the limitations of their work, particularly regarding efficiency, simplification, and model-dependency in evaluation.

---

> ### Author Rebuttal · Authors · 2024-08-06
>
> We are with great pleasure to see your strong recommendation of our work, thank you! We make the following further improvements and discussions to our paper corresponding to your insightful questions and suggestions,
>
> ## **Student model selection**
>
> We select DeBERTa as the student model because the assigned tasks (Relevance and NLI) are classic NLU tasks, on which encoder models (BERT, RoBERTa, etc.) show strong performance (referring to the GLUE benchmark). To further justify the DeBERTa model used in our experiments, we incorporate a comparison among the student models about their distillation efficiency and evaluation efficiency in the **Implementation Justification** section in **Author Rebuttal by Authors**. The results in the **4th table in Author Rebuttal by Authors** show DeBERTa can rival LLMs like Gemma in the distillation performance while significantly boosting the evaluation efficiency. Thus, we select DeBERTa as the student model for experiment efficiency.
>
> ## **Incorporating human annotation**
>
> Human annotations are helpful to better align the evaluation with humans, but NLI and similarity are classic and deterministic NLP tasks, so we use an off-the-shelf model like the state-of-the-art LLM, GPT-4. As the score is decomposed to simple NLP tasks, GPT-4 should show a convincing performance. In future works, it will still be nice to devote human effort to correcting the potential mistakes in the annotations of GPT-4.
>
> ## **More comprehensive score**
>
> Our APC score can be certainly extended to a more comprehensive metric by including statements other than persona statements in the constraint satisfaction problem. In the current Appendix I, we give an example of how to inject preference from other dimensions (protective experience, which forbids characters from knowing inexperienced stuff). For those explicit dimensions that can be written down as texts, they can be likely incorporated. For implicit dimensions, like "consistent with dialogues in the training set" (in dialogue-driven role-playing), we have to use more implicit scorers to judge whether the response is consistent, introducing other types of constraint into the system since extra materials - dialogues - are involved in.
>
> ## **Discussion about addressing weakness**
>
> Thank you for pointing out the weaknesses of the current formulation of our APC score. However, we have discussed them in the limitation section together with potential ways to address them in future work. Here we can deepen the discussion about how we can get rid of the limitations.
>
> **Simplification** can be handled by taking the user's preference $p$ into the constraint satisfaction problem, which transforms $Rel(q,s)$ into $Rel(p,q,s)$ (sometimes people can define what is relevant). This can reweigh the importance of each constraint according to the dynamic preference of the user, thus addressing the simplification limitation of APC.
>
> **Model dependency** requires to be addressed by incorporating human annotations, which can be acquired by directly annotating or correcting the annotations from state-of-the-art LLMs like GPT-4.
>
> **Other evaluation dimension** can be addressed by explicitly or implicitly incorporating more constraints from the role-playing system designer as illustrated in the **More comprehensive score** section of this rebuttal.

---

### Official Review · Reviewer_5oeH · 2024-07-13

**Soundness:** 3
**Presentation:** 3
**Contribution:** 4
**Rating:** 6
**Confidence:** 3

**Summary:**

The paper proposes a new evaluation metric, delta APC score, which uses constraint satisfaction inspiration to tackle the evaluation of faithfulness to persona descriptions. Then, evaluations are conducted on the experience upload, RAG, and long-context memory approaches. To do this, 3 personas are created with varying statements counts, and these three approaches are compared across the 3 personas.
Next, the authors introduce delta APC as a reliable reward component for DPO. This evaluation is conducted on famous figures with more statements. Authors find APC to work well with existing DPO for faithfulness.

**Strengths:**

Originality: The formulation of faithfulness as constraint satisfaction is novel and interesting.

Quality: The implementations are solid and evaluations are done on three methods, EU, RAG, and LCM. Mathematical sections are solid and well-appreciated. Each claim made in the abstract has evidence in the experiments to support it. Scaling rules is a nice additional touch.

Clarity: Writing is mostly clear and easy to follow. It might be good to add some more detail when introducing constraint satisfaction for readers with a LLM background.

Significance: Faithfulness is an important attribute for persona simulations. This is the first benchmark that uses constraint satisfaction in such evaluations.

**Weaknesses:**

The main weakness of the paper is that the number of profiles being evaluated on for the experiments is too few and feels non-generalizable. An addition of more profiles (along with some sort of notion of demographic-level population representation) would help with this. This is particularly the case with the Alice Bob set of evaluations.

More generally, the evaluations seem a bit on the lower side, so any additions that the authors come up with (or that other reviewers suggest) would help.

Small notes:
Some acronyms (PRP, APC) are used a bit too much that they break reading flow.

The paper is missing some citations to past works such as [1].

[1] Park, Joon Sung, et al. "Generative agents: Interactive simulacra of human behavior." Proceedings of the 36th annual acm symposium on user interface software and technology. 2023.

**Questions:**

My main question for the authors is: what benefits does formulating persona faithfulness as a constraint satisfaction problem bring as benefit compared to other approaches? The mathematical formulation is nice, but does a similar formulation exist for other approaches? It does not seem immediately clear in the text.

In table 1, CPO is evaluated on APC, and it seems intuitive that it would achieve better results because these are objectives that it is supposed to perform well on. Are there any other metrics that you can evaluate CPO on that also measure faithfulness? This would make its contribution more convincing.

People are generally not consistent and static. They will often change based on their mood, learning, and shift their opinion. How should a system such as APC address such phenomena?

**Limitations:**

Limitations are mentioned.

---

> ### Author Rebuttal · Authors · 2024-08-06
>
> We appreciate your positive attitude towards our work and the effort to provide valuable suggestions to further polish our work. To address your concerns, we will add the following extra experiments and further clarification to the final version of our paper.
>
> ## **More demographic-level representative profiles**
> We add 9 new original characters to our experiments, referring to the **Experiment Comprehensiveness** section in **Author Rebuttal by Authors**. These characters are carefully designed to be with different spatial (Asian, European, African, etc.) and temporal (past, present, future) conditions. They also include fantasy characters like a time-traveling scientist, thus broadening the generalizability verified from the experiment.
>
> The results of these new diverse characters **(1st and 2nd tables in Author Rebuttal by Authors)** achieve a consistent conclusion with the existing experiments, thus further strengthening the generalizability of our discovery. We will further include human evaluation for these new characters in the final version of our paper.
>
> ## **Why formulating persona faithfulness as a constraint satisfaction problem?**
>
> Intuitively, "faithfulness" can be viewed as "obeying pre-defined rules", which can be naturally connected with constraint satisfaction problems. In practice, viewing faithfulness as constraint satisfaction helps us to break down the coarse-grained concept of faithfulness into simple constraints for satisfaction like relevance and NLI, which are simple and classic NLU tasks with a nice explainability. Also, it helps to better compare role-playing methods such as "method A is better than method B" because "A satisfies more constraints than B" rather than "LLM says A is more faithful than B". To the best of our knowledge, we are the first to decompose the profile into persona statements for fine-grained faithfulness evaluation, which is an important novelty of our work. We will include the clarification above to further emphasize our contribution, thank you!
>
> ## **Other faithfulness evaluator**
>
> Currently, we contain human evaluation and a case study to re-verify the benefit of optimizing towards our proposed APC score metric. For automated metrics, the existing evaluation of attributes like faithfulness in role-playing methods is based on prompting LLMs for a coarse-grained evaluation. Thus, it will be hard to find another metric (other than human evaluation) that can double-check the faithfulness of the role-playing methods. Instead, we can compare our fine-grained APC score with the coarse-grained metric on their consistency with human evaluation, which is shown in the **Metric Justification** section in **Author Rebuttal by Authors**. The results **(3rd table in Author Rebuttal by Authors)** verify the fine-grained APC score is more consistent with human evaluation and stable to the number of persona statements. Thus, we can also say optimizing towards a high-quality objective can naturally better improve the global faithfulness of role-playing methods.
>
> ## **Handling dynamic preferences of people**
>
> You point out an important point to evaluate the deployed role-playing systems. From our constraint satisfaction view, we should incorporate the preference $p$ of the user into the constraint, which transforms $Rel(q,s)$ into $Rel(p,q,s)$ (sometimes people can define what is relevant) and $NLI(q,s,r)$ into $NLI(p,q,s,r)$ (people might not care about some constraints are satisfied). While people's preference makes the evaluation more complicated, it still fits into the constraint satisfaction framework with some modification. We can use this modified constraint to handle the dynamic preferences of people.
>
> ## **Writing and Citation**
>
> Thank you for pointing out the weaknesses in our writing, we will reduce the number of acronyms to avoid breaking the reading flow. We will also cite more related role-playing past works like [1] to strengthen the connection with other works.

---

> > ### Comment · Reviewer_5oeH · 2024-08-12
> >
> > Thank you for your response and clarifications. I have updated my score accordingly and vote for acceptance.
> >
> > Regarding the last point about people shifting their opinion - here I was referring to how people that you are trying to simulate can change their opinions. Is this also able to be adapted into the constraint satisfaction framework?

---

> > > ### Author Response · Authors · 2024-08-13
> > > **Response to Reviewer 5oeH**
> > >
> > > Thank you for the clarification on the question! For scenarios that the simulated character might shift the persona statement pool, we will discuss the application of APC in both quantification and optimization.
> > >
> > > **Quantification: ** Our APC scheme can be directly applied to evaluate persona-shifting characters because of its plug-and-play property. We can use the trained discriminators (relevance and NLI) with a new pool of persona statements after increase, decrease, or modification. As our APC scoring system estimates the number of constraints satisfied, it can be aggregated among different persona statement pools. So even in a dialogue that the character shifts its persona in the middle, we can evaluate the faithfulness in the full dialogue by evaluating with different statement pools before and after the shifting.
> > >
> > > **Optimization: ** The current formulation of the DPO optimization might not be directly applied to handle shifting persona statement pool since the reward model is based on a static preference model. Fortunately, we can adapt the reward model to a dynamic one by taking a group of dynamic persona statements in the input side $APC(r, S_{dynamic}|q, S_{dynamic}, S_{static})$ (need to have a designed area to input these shifting persona statements so the optimization will encourage the AI character to follow the changeable input $S_{dynamic}$). For instance, $S_{dynamic}$ can be “Alice is happy”, “Alice is sad”, … The character LLM also needs to be adapted to have a input area for $S_{dynamic}$. One possible limitation of this scheme is when we want to maintain a large pool of dynamic statements, it will challenge the LLM’s long-context ability as shown in the LCM part in the current paper.
> > >
> > > We hope the explanation above addresses your concern, thank you!

---

### Author Rebuttal · Authors · 2024-08-06

We are sincerely thankful for all reviewers' positive feedback and insightful suggestions to improve the quality of our work. We are glad to address your concerns with further clarification and more experiment results for support. Here we include the responses to weaknesses and questions mentioned by multiple reviewers as a reference for rebuttals.

## **Experiment Comprehensiveness**

We completely agree with the importance of more characters with demographic-level population representation to solidify our conclusion. Thus, we add experiments (as in Table 1) for 9 new carefully designed characters each with 30 original persona statements in their profile for experiment efficiency, which include:

- Alex: An African American baseball player
- Isabella: An Italian traveling cook
- Takayoshi: A Japanese game developer

(Characters in the past)

- Ousmane: A rich gold mine owner of the Malian Empire in the 1300s
- Jones: A young British worker in the Victorian Era
- Zhe: A Chinese poet in the Tang Dynasty

(Characters in the fantasy)

- Crossan: A time-traveling scientist
- Betty: A pet cat who can talk with ghosts
- X: An alien space traveler and photographer

These characters can better represent people with different spatial and temporal backgrounds and even cover non-human characters from the fantasy world. The specific persona statements will be attached to the final version of our paper.

The results are shown as follows,

***(DeBERTa Evaluator)***

|Character|Alex|Isabella|Takayoshi|Ousmane|Jones|Zhe|Crossan|Betty|X|
|---|---|---|---|---|---|---|---|---|---|
|Vanilla|0.5|0.8|0.6|0.3|0.9|1.1|0.3|0.3|0.7|
|EU|1.8|2.8|2.0|1.4|0.7|3.8|2.0|1.2|5.2|
|LCM|7.1|7.4|6.5|4.5|6.2|5.2|2.2|2.8|8.1|
|RAG|7.6|8.1|6.9|3.0|6.6|5.8|1.8|3.2|7.5|
|EU w/ CPO|5.3|6.1|5.7|3.6|4.8|4.9|3.1|2.9|7.9|
|LCM w/ CPO|7.5|7.7|7.0|**4.8**|6.2|5.4|**4.5**|3.9|8.2|
|RAG w/ CPO|**7.9**|**8.2**|**7.4**|3.9|**7.5**|**6.9**|2.5|**4.6**|**8.9**|

***(GPT-4 Evaluator)***

|Character|Alex|Isabella|Takayoshi|Ousmane|Jones|Zhe|Crossan|Betty|X|
|---|---|---|---|---|---|---|---|---|---|
|Vanilla|0.2|0.1|-0.2|0.5|0.2|0.2|0.4|0.1|0.8|
|EU|1.4|1.8|3.0|0.5|1.3|6.4|1.2|0.3|7.4|
|LCM|3.1|8.6|5.6|4.1|7.4|3.4|2.1|1.6|11.3|
|RAG|3.3|7.8|6.1|1.6|8.1|4.3|2.7|2.2|10.1|
|EU w/ CPO|2.7|5.6|5.9|3.0|4.7|7.1|2.2|1.5|9.5|
|LCM w/ CPO|3.2|9.8|8.1|**4.6**|8.2|**7.8**|**4.0**|2.3|12.1|
|RAG w/ CPO|**4.8**|**10.0**|**9.8**|2.0|**8.3**|7.3|2.9|**3.1**|**14.6**|

The result is consistent with the conclusions in our paper that RAG outperforms other methods and w/ APC-based DPO generally RAG shows the best performance. We will attach this table and specific profiles of these 9 new characters to the appendix of our paper. We will also include human evaluation for these characters in the final version of our paper.

## **Metric Justification**

To better justify selecting our APC score and also support the claim that the fine-grained APC score has the advantage over coarse-grained metrics, we add a coarse-grained metric as the baseline. We directly prompt GPT-4 with the criterion used for human evaluation shown in Appendix E. We also distill this scoring ability (following the same scenario as APC) to DeBERTa to check whether the efficiency can be boosted. We evaluate the Spearman correlation between the metric and the human evaluation of the 7 role-playing methods on the 3 human-evaluated characters.

|Character (#Persona Statement)|Alice (8)|Bob (19)|Eve (30)|
|---|---|---|---|
|Coarse-grained Score (GPT-4)|92.42|86.27|81.40|
|APC Score (GPT-4)|97.18|99.10|99.10|
|Coarse-grained Score (DeBERTa)|81.40|69.91|54.57|
|APC Score (DeBERTa)|88.61|95.50|99.10|

The results verify that 1) Fine-grained APC score shows better consistency with human evaluation. 2) The fine-grained APC score is stable to the number of persona statements while the coarse-grained score degrades with the increase of persona statements. 3) The coarse-grained evaluating ability is harder to be distilled into smaller models for efficiency boosting. Based on case checking, we find an underlying issue of the coarse-grained metric is the LLM will assign a high score to a response once it contains some correct information, ignoring the missing important information (active constraint) and occasionally conflictions (passive constraint). We will further include the evaluation of the new 9 characters together with the human evaluation in the final version of our paper.

## **Implementation Justification**

We select DeBERTa as the student model to distill from GPT-4 because small encoders (BERT, RoBERTa, etc.) show promising performance on relevance and NLI, which are classic NLU tasks in the GLUE benchmark. Among encoders, DeBERTa (DeBERTa-v3-large) is a state-of-the-art model that shows strong performance after fine-tuned on NLU tasks. To further verify DeBERTa as a proficient student model, we add an analysis of the in-domain/out-of-domain performance and the efficiency of different base models for distillation.

|Task|Rel. (In-domain)|Rel. (Out-of-domain)|NLI (In-domain)|NLI (Out-of-domain)|Efficiency|
|---|---|---|---|---|---|
|DeBERTa (Base)|92.46|89.90|89.72|87.80|409.6it/s|
|DeBERTa (Large)|94.04|92.10|93.46|91.50|150.8it/s|
|Gemma-1.1-it (2b)|94.25|92.50|93.68|91.80|26.4it/s|

The in-domain test set (1697 instances for Relevance, 3773 instances for NLI) is the 20% split of the characters (Beethoven, Newton, Socrates) that build the training set (6787 instances for Relevance, 15092 instances for NLI). The out-of-domain test set samples 1000 cases from other characters. The results show DeBERTa-V3-Large (300M) shows a comparative performance with a 2B Gemma model, while is about 6 times faster, which justifies DeBERTa to be a strong student model. The out-of-domain performance is generally high, which indicates the generalizability to other characters. Finally, an extra discovery is that DeBERTa-v3-base (100M) can further significantly boost efficiency with some trade-offs in accuracy.

---

### Decision · Program_Chairs · 2024-09-25

**Decision:**

Accept (poster)

**Comment:**

The paper introduces a novel Active-Passive-Constraint (APC) score to evaluate persona-driven role-playing (PRP) faithfulness. APC score correlates with human evaluation and enhances AI character responses using direct preference optimization (DPO).

The reviews have raised questions around limitations on generalizability of the evaluation, scaling up and the benefits of using a constraint satisfaction approach to persona faithfulness in AI characters and how can a system like APC account for the dynamic nature of human personalities and behaviors. The authors have answered all the questions and updated the submission with additional experiments.

The authors recently added additional experiment results reinforcing previous questions including overfitting to specific characters, and to support the lack of human evaluation for the new metric, they introduced a new coarse-grained metric (though no human eval is conducted on the new metric)